# Biobased and biodegradable films exhibiting circularly polarized room temperature phosphorescence

Mengnan Cao[1], Yiran Ren[2], Yue Wu [2] ✉, Jingjie Shen[1], Shujun Li [1] ✉, Zhen-Qiang Yu [2], Shouxin Liu[1], Jian Li[1], Orlando J. Rojas [3,4,5] ✉ & Zhijun Chen [1] ✉

There is interest in developing sustainable materials displaying circularly polarized room-temperature phosphorescence, which have been scarcely reported. Here, we introduce biobased thin films exhibiting circularly polarized luminescence with simultaneous room-temperature phosphorescence. For this purpose, phosphorescence-active lignosulfonate biomolecules are co-assembled with cellulose nanocrystals in a chiral construct. The lignosulfonate is shown to capture the chirality generated by cellulose nanocrystals within the films, emitting circularly polarized phosphorescence with a 0.21 dissymmetry factor and 103 ms phosphorescence lifetime. By contrast with most organic phosphorescence materials, this chiral-phosphorescent system possesses phosphorescence stability, with no significant recession under extreme chemical environments. Meanwhile, the luminescent films resist water and humid environments but are fully biodegradable (16 days) in soil conditions. The introduced bio-based, environmentally-friendly circularly polarized phosphorescence system is expected to open many opportunities, as demonstrated here for information processing and anti-counterfeiting.

Organic room temperature phosphorescence (RTP) materials possess many attributes, such as large Stokes shifts, long-lived emission and favorable processability[1–6], offering a great potential as building blocks of multifunctional optical materials[7]. Amongst RTP materials, those with circularly polarized luminescence (CPL) have received particular attention because of their technological impact, for instance, in 3D optical displays[8–11], information storage[12–14], asymmetric catalysis[15,16], and anti-counterfeiting devices[17–19].

Generally, the two main routes used to produce CPL materials[20–23] include the use of building blocks exhibiting chirality as well as RTP emission[24,25] and the co-assembly of an achiral RTP emitter with a chiral material, creating circularly polarized room-temperature phosphorescence (CP-RTP)[26–28]. Guided by these strategies and considering their sustainability and low-carbon intensity, renewable resources have captured the interest as precursors of CPL materials[29–31]. Particularly, aqueous suspensions of cellulose nanocrystals, CNC ( ~ 100–500 nm by ~ 5–20 nm)[32,33] have been shown to form chiral nematic mesophases and, upon drying, lead to solid films that reflect the initial structure (e.g., with CNC aligned in a chiral helix). Hence, CNC is an excellent candidate as chiral host to achieve CPL.

To date, luminophores such as molecular chromophores, carbon dots, etc. have been incorporated into chiral nematic CNC films to obtain CPL emission[34,35]. Nevertheless, most of the reported luminophores require multi-step preparation protocols

[1]Key Laboratory of Bio-based Material Science and Technology of Ministry of Education, Northeast Forestry University, Harbin 150040, China. [2]College of Chemistry and Environmental Engineering, Shenzhen University, Shenzhen 518071, China. [3]Bioproducts Institute, Department of Chemical & Biological Engineering, University of British Columbia, Vancouver, British Columbia, Vancouver, BC V6T 1Z3, Canada. [4]Department of Chemistry, University of British Columbia, Vancouver, BC V6T 1Z1, Canada. [5]Department of Wood Science, University of British Columbia, Vancouver, BC V6T 1Z4, Canada. ✉e-mail: wuyue@szu.edu.cn; lishujun@nefu.edu.cn; orlando.rojas@ubc.ca; chenzhijun@nefu.edu.cn

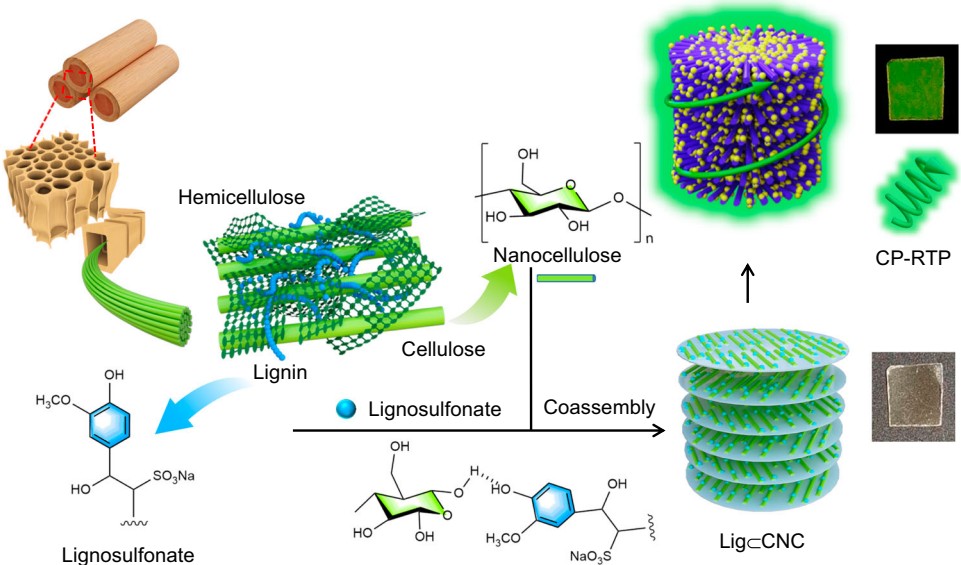

**Fig. 1 | Schematic illustration for the preparation of sustainable circularly polarized room-temperature phosphorescence (CP-RTP).** Chiral cellulose nanocrystals (CNC) and a phosphorescence-active technical lignosulfonate (Lig) cooperatively assemble into phosphorescent chiroptical thin films.

and still rely on unsustainable fossil carbon components, hindering CNC appeal as a biobased material. Such facts highlight the unmet objective of developing systems from emitters that fulfil the demands of performance, facile processing, biodegradability, sustainability and cost effectiveness.

The properties of lignosulfonates make them a viable option in biobased CPL[36] especially considering that they are widely available as a product of wood digestion in the forest products industries[37,38]. Moreover, lignosulfonates are commercially available and can be solubilized in water, facilitating processing with CNC. Chemically, lignosulfonates are composed of aromatic units that show photoactivity[39], such as, fluorescence[40], phosphorescence[41], photothermal[42,43] and photocatalytic effects[44]. Nevertheless, few efforts have reported the successful use of lignosulfonates for CPL materials.

In this work, we introduce a rational strategy that combines chiral CNC with RTP-active lignosulfonates by a co-assembly process (Fig. 1). Lig⊂CNC films are demonstrated to produce circularly polarized room-temperature phosphorescence (CP-RTP), bestowing the following remarkable combination of features: (i) high-dissymmetry CP-RTP ($g_{lum}$ = 0.21) originated from the chiral ordered co-assembly, (ii) remarkable RTP enhancement (57-fold) compared to lignosulfonate-free CNC films, and (iii) high chemical stability with simultaneous biodegradability in soil conditions.

## Results

### Room-temperature phosphorescence generated by an all-biobased material

The photo-physical properties of CNC were firstly investigated in the form of thin films. As illustrated in Fig. 2a, CNC films were colorless and emitted cyan luminescence under 365 nm UV light irradiation, with a maximum luminescent wavelength at 480 nm. Though no afterglow by the naked eye when the UV light was turned off was observed, the lifetime of CNC emission was 9.12 ms at 520 nm and 298 K (Supplementary Fig. 1). These observations suggest that the emission peak at 535 nm can be assigned to phosphorescence. The CNC thin film showed a typical chiral assembly with fingerprint textures, as shown in the polarizing optical microscope (POM) images acquired at room temperature (Supplementary Fig. 2). CNC and lignosulfonate were assembled cooperatively as an all-biobased and

bright luminogen, Lig⊂CNC. Compared to the CNC film, the Lig⊂CNC system emitted stronger cyan luminescence excited by 365 nm UV light. Time-resolved emission decay 2D spectra showed that 0.25 wt% Lig⊂CNC film exhibited a long-lasting and stable phosphorescence with 103 ms lifetime at 520 nm and 75.22 ms lifetime at 480 nm (Fig. 2b and Supplementary Fig. 3), and green afterglow was observed by the naked eye, lasting for 1 s when UV illumination was turned off (Fig. 2a).

As expected, by combining CNC with lignosulfonate (0.25 wt%) and upon excitation at 365 nm, the luminescence and RTP intensity of the Lig⊂CNC film were shown to be *ca.* 20-fold and 57-fold higher compared to that of the unmodified CNC film, respectively (Fig. 2c, d). This result highlights the role of lignosulfonate toward the enhancement of light emission. To further understand such effect, the interactions between CNC and lignosulfonate were considered: both the CNC and lignosulfonate are negatively charged (zeta potential of −42 and −36 mV, respectively), indicating a minor role of electrostatic interactions (Supplementary Fig. 4). Meanwhile, Fourier infrared spectroscopy (FT-IR) showed that compared to the neat lignosulfonate and CNC, the characteristic -O-H tensile vibration signal (3681–2987 cm$^{-1}$ range) of Lig⊂CNC was significantly enhanced (Supplementary Fig. 5). The results point to the possibility of hydrogen bonding between lignosulfonates and CNC[45] which restricts the molecular vibration of lignosulfonates and enhances RTP emission[46]. To further confirm the generality of hydrogen bonding effect in RTP enhancement, two other polymers were combined with the lignosulfonate, polyvinyl alcohol (PVA) and polystyrene (PS). As shown in Supplementary Fig. 6, Lig⊂PS, which is free of hydrogen bonding between the two components, displayed very weak phosphorescence (with 2.95 ms lifetime). Conversely, the co-assembled Lig⊂PVA clearly exhibited enhanced phosphorescence (168.63 ms lifetime), given the effect of hydrogen bonding. These results support the hypothesis that hydrogen bonding between CNC and the lignosulfonate contributes to the enhancement of the RTP property in the Lig⊂CNC system.

Notably, the luminescence intensity at 480 nm dropped gradually as the mass fraction of lignosulfonates in the film ($f_{Lig}$) was increased ($f_{Lig}$, between 0.5 to 2%, Fig. 2c). In agreement with the photoluminescence spectra, Lig⊂CNC films of different $f_{Lig}$ possessed a similar RTP profile at 535 nm, with an optimal RTP at

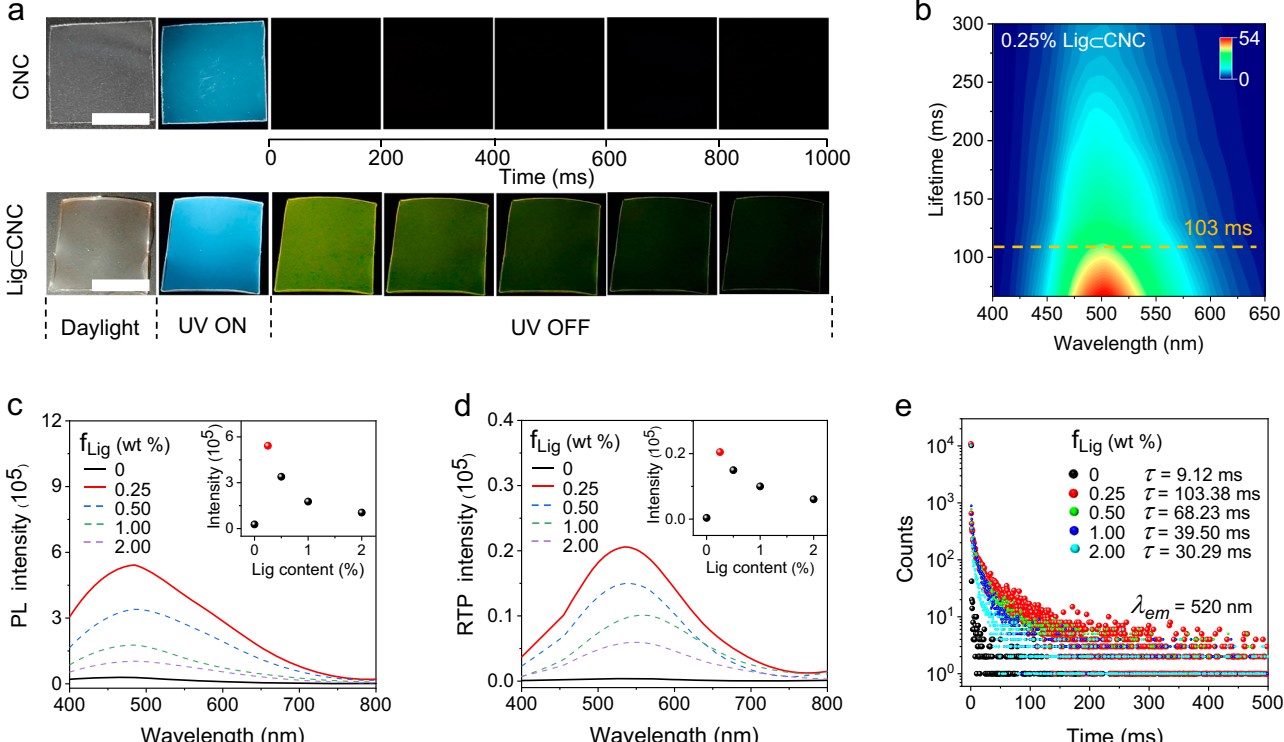

**Fig. 2 | Photophysical properties of assembled CNC and co-assembled Lig⊂CNC.**
**a** Photos showing time-dependent photoluminescence of CNC and co-assembled Lig⊂CNC films. Scale bar = 1 cm. **b** Evolution of 2D luminescent spectra of Lig⊂CNC in air. Changes in luminescence (**c**) and phosphorescence (**d**) spectra of a CNC thin film loaded with lignosulfonate (Lig) at given mass fractions, $f_{Lig}$ = 0 wt % (black profile), $f_{Lig}$ = 0.25 wt % (red line), $f_{Lig}$ = 0.50 wt % (blue line), $f_{Lig}$ = 1.00 wt % (green line), $f_{Lig}$ = 2.00 wt % (purple profile) ($\lambda_{ex}$ = 365 nm, 25 °C). Insets in (**c**) correspond to the changes in photoluminescence at 480 nm and (**d**) phosphorescence at 535 nm. In the experiment, the concentration of CNC was kept constant and the concentration of lignosulfonate was varied (between 0 and 2%, as noted). **e** Luminescence decay curves of Lig⊂CNC at different $f_{Lig}$.

$f_{Lig}$ = 0.25% (Fig. 2d). The finding of a co-assembly for maximum photoluminescence indicates that lignosulfonates and CNC formed a RTP co-assembly under conflicting effects. For instance, it is likely that excess lignosulfonate (>0.25%) disrupted the co-assembly of the system, leading to RTP efficiency attenuation. The lifetime measured for Lig⊂CNC films with $f_{Lig}$ = 0, 0.25%, 0.50%, 1.0%, and 2.0% corresponded to 9.12, 103.38, 68.23, 39.50, and 30.29 ms, respectively (Fig. 2e), further confirming an optimal co-assembly at $f_{Lig}$ = 0.25% (103.38 ms lifetime).

The effect of temperature on the phosphorescence lifetime of Lig⊂CNC films was also investigated: the luminescence lifetime of Lig⊂CNC decreased as the temperature increased from 77 K to 300 K (Supplementary Fig. 7), in keeping with the expected reduction in phosphorescence at elevated temperatures given the contribution of non-radiative decay. Moreover, the afterglow RTP thin film was found to be humidity-sensitive. In particular, the phosphorescence lifetime dropped from ~102.28 ms to ~ 22.82 ms when the humidity increased from 10% to 90% (Supplementary Fig. 8). Interestingly, this quenched lifetime was recovered by drying the Lig⊂CNC film. The humidity-drying RTP lifetime was stable and was recovered by drying, with no hysteresis, as confirmed after four cycles (Supplementary Fig. 9). Finally, the phosphorescence of Lig⊂CNC was sensitive to $O_2$: the lifetime at 520 nm dropped to ~45.61 ms under 90% fraction of $O_2$ environment (Supplementary Fig. 10). As will be shown later, however, the performance of Lig⊂CNC film was not affected under harsh chemical (solvent) environments.

## Chirality induces CP-RTP rather than selected transmission

CNC is a nanoparticle that presents a twist and its chiral properties in the context of Lig⊂CNC films were further examined. Scanning electron microscopy (SEM) images of CNC films revealed a chiral nematic structure with helical pitch polydispersity (Supplementary Fig. 11). As shown in Fig. 3a, b, similar nematic-like domains were observed in Lig⊂CNC films ($f_{Lig}$=0.25 wt%). However, when the lignin loading increased to 1% or 2%, the chiral co-assembly was disrupted (Supplementary Fig. 12), pointing to the choice of 0.25 wt% lignosulfonate as appropriate in the co-assembled Lig⊂CNC. Beyond regular luminescence, the chiral transmission emerged from the synchronous effects of adsorbed lignosulfonate in Lig⊂CNC. Since CNC thin films possessed a typical chiral assembly, we assessed the chiroptical activity by circular dichroism (CD), and circularly polarized luminescence (CPL). CD and CPL were used to investigate the chirality transfer from the CNC thin film to the lignosulfonate with the CD sign indicating the chirality transmission from the view of ground-state supramolecular co-assembly. Remarkably, the CNC thin film possessed a strong CD signal with a maximum peak at 460 nm, Fig. 3c, d. The corresponding dissymmetry factor of absorption ($g_{abs}$) reached a value of 0.9. To the best of our knowledge, this is the highest absorption dissymmetry for CNC. In the presence of lignosulfonate ($f_{Lig}$=0.25 wt%) the co-assembled Lig⊂CNC film displayed a similar CD sign, confirming that the CNC-based chiroptical property was preserved.

Beyond the ground-state chiroptical property (CD), we employed CPL to investigate the excited-state chiroptical effect. Indeed, CPL provides information on the macroscopic chirality of chiral luminescent or phosphorescent dyes in the excited state. The primary criterion for appraising CPL is to measure the luminescence dissymmetry factor, $g_{lum} = 2(I_L - I_R)/(I_L + I_R)$, which denotes the excited-state luminescence difference of left-handed (L) and right-handed (R) circularly polarized light[47–49]. For CPL, a high $g_{lum}$ value is an important measure of the chiroptical property. As shown in Fig. 3e, a strong negative CPL signal

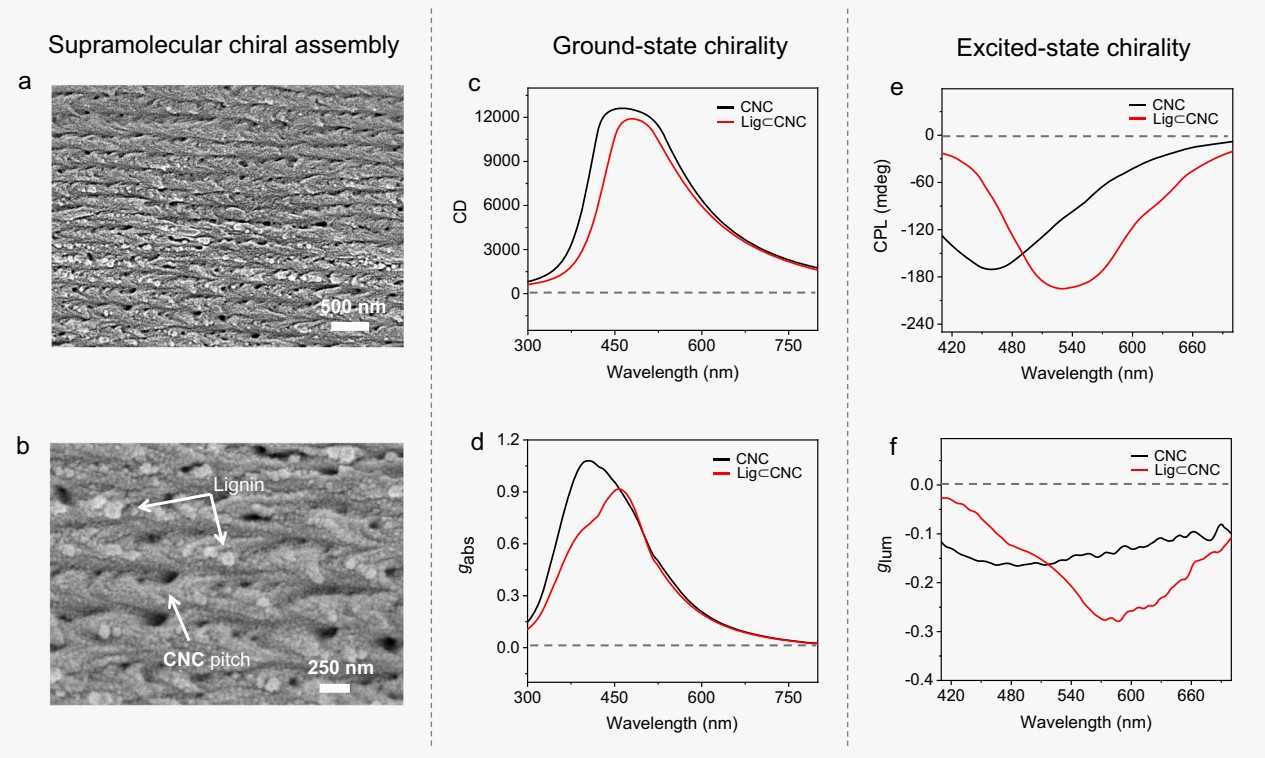

**Fig. 3 | Chiroptical properties of films produced from CNC and co-assembled Lig⊂CNC. a, b** Scanning electron microscope (SEM) images of Lig⊂CNC displays predominantly chiral order with a helical pitch. **c** Circular dichroism (CD) and (**d**) dissymmetry factor of absorption ($g_{abs}$) value spectra for CNC (black line) and Lig⊂CNC (red line) in the thin-film state. **e** Circularly polarized luminescence (CPL) and (**f**) luminescence dissymmetry factor ($g_{lum}$) spectra for CNC (black line) and Lig⊂CNC (red line) thin films excited at 365 nm light.

was observed for a CNC thin film at 460 nm, fully consistent with its luminescence wavelength. Furthermore, the calculated value of $|g_{lum}|$ of CNC was ~$1.6 \times 10^{-1}$ (Fig. 3f) as a thin film, compared to $10^{-4}$–$10^{-2}$ reported in previous work[50–52].

Next, we incorporated lignosulfonate into the CNC thin film and showed that the CPL peak of the Lig⊂CNC shifted from 460 to 540 nm, upon 365 nm light excitation. This belongs to the RTP range of Lig⊂CNC, directly displaying CP-RTP. Similarly, the maximum $g_{lum}$ peak shifted from 460 to 570 nm in agreement with CPL, which is also located in the zone of RTP (Fig. 3e). Within the intermolecular system, the chirality transfer between chiral CNC and achiral lignosulfonate was successfully achieved in the thin films. Interestingly, the maximum calculated $|g_{lum}|$ of Lig⊂CNC reached a value of 0.21, which is considerably higher than that of the neat CNC thin film ($|g_{lum}|$ = 0.16). This is indicative of a highly efficient chirality transmission within the intermolecular co-assembly mode in Lig⊂CNC (Fig. 3f). To illustrate such CPL amplification effect, we found that the lignin absorption could match the CNC luminescence (Supplementary Fig. 13), and conducted two luminescence decay experiments on CNC and Lig⊂CNC. As shown in Supplementary Fig. 14, the luminescence lifetime of neat CNC at 440 nm was 80.19 ms and the luminescence quantum yield was 0.62%; whereas the lifetime of co-assembled Lig⊂CNC was shortened to 49.91 ms, and the corresponding luminescence quantum yield was 3.75%. These results indicate the effect of a Förster resonance energy transfer (FRET) process from CNC to lignin, within the intermolecular chiral co-assembly system. The FRET efficiency was calculated as 37.8%[53], thus resulting in a CPL $|g_{lum}|$ amplification (0.16 → 0.21) from CNC to Lig⊂CNC, which has been verified in previous works[23,54–56]. Coupled with the RTP property, Lig⊂CNC showed a specific CP-RTP function.

To confirm the high $|g_{lum}|$ value of Lig⊂CNC from intermolecular chiral FRET rather than selected transmission, a light intensity detection experiment was carried out. As shown in Fig. 4a, we used a 360°-rotation light intensity determination diagram: i) Converting 532 nm laser into linearly polarized light by using a fixed polarizer, ii) passing through the sample (Lig⊂CNC thin film) and, iii) determining the light intensity at different angles after the 360° polarizer rotation, where the polar angle ω stands for the transmission angle of the polarizer and the radius r stands for the transmittance.

The laser characteristic was evaluated optically by analyzing the azimuthal angle-dependent transmission with an "8" profile with the strongest and weakest light intensity at 135/315° and 45/225°, respectively, clearly ruling out selected transmission/reflection-dependent CPL (Fig. 4b). To further verify chirality-induced CP-RTP, a Bragg reflection test was carried out. As shown in Fig. 4c, the Bragg reflection of CNC and Lig⊂CNC thin films were both away from the visible light wavelength region, and nearly no reflection color was observed, which rule out thin film stacks and selected reflection. According to the optical setup in Fig. 4a, we also conducted a selective transmission test for a chiral liquid crystal (2.33 w% R5011 in TEB300), used as a reference, which possess a reflection band from 500 to 550 nm (Supplementary Fig. 15). The laser characteristic at 532 nm was evaluated optically by analyzing the azimuthal angle-dependent transmission with a round curve, demonstrating a typical selective transmission, as shown in Supplementary Fig. 16.

## Information processing based on CP-RTP
Having established the characteristics of the CP-RTP, we turned our attention to a chiroptical application. The excellent dynamic chiral phosphorescence properties of Lig⊂CNC mean that such films can be used in optical devices, for instance, for information encryption. Because the Lig⊂CNC possesses both fluorescence and phosphorescence, we explored the possibility of generating an anticounterfeiting pattern on paper, for instance by using imprint lithography (Fig. 5a).

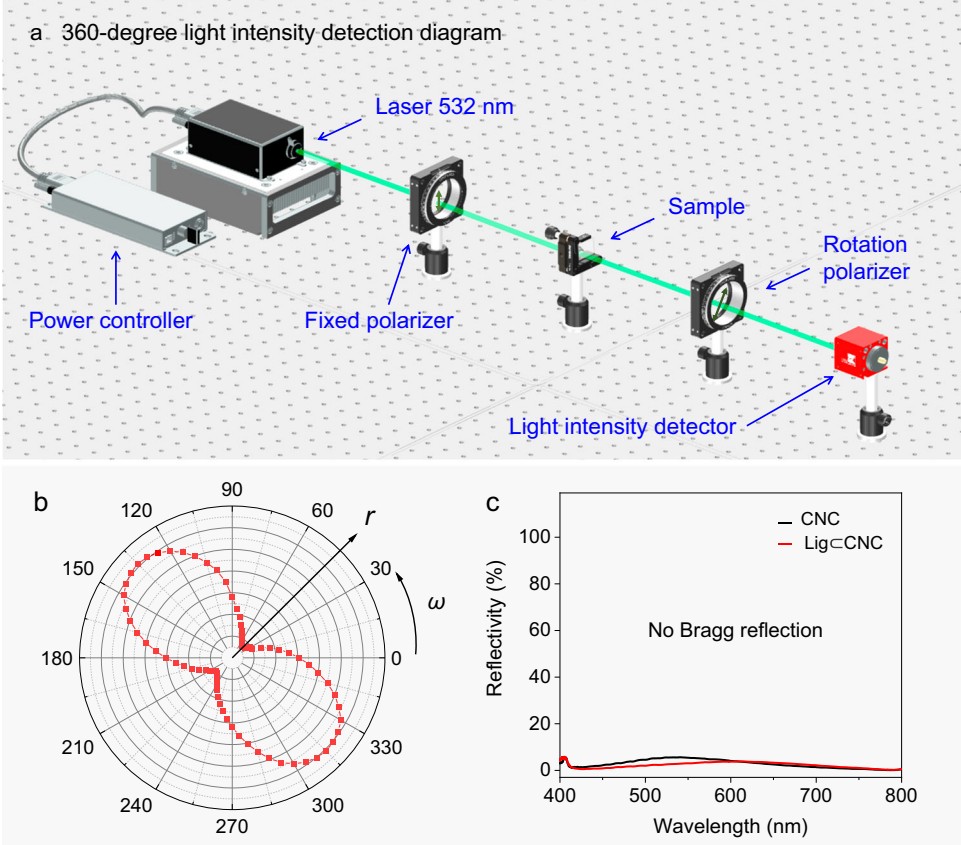

**Fig. 4 | Verifying the chirality-induced CP-RTP. a** Optical setup of azimuthal angle-dependent transmission. **b** Transmission intensity of linearly polarized light at 532 nm across a Lig⊂CNC thin film. **c** The weak reflection wavelength band of CNC (black line) and Lig⊂CNC (red line) thin films indicating nearly null Bragg reflection ranging from 400 to 800 nm.

Sixteen letters were imprinted on CNC and Lig⊂CNC films, except for C, P, and L in the latter case. As illustrated in Fig. 5a, a white paper showed the sixteen letters with cyan luminescence under 365 nm UV light irradiation. When light at 365 nm was turned off, only three letters (C, P, and L) appeared in green. These characteristics indicated the promise for data encryption.

The CD and CPL spectra of Lig⊂PVA film were acquired and Supplementary Fig. 17 depicts nearly null CD and CPL signals, which confirms chiral CP-RTP coming from CNC. Thus, we conceived a display with anti-counterfeiting pattern 888 using different phosphors (Fig. 5b). The photoactivation results are shown in Fig. 5b: (i) Under 365 nm UV light excitation, the luminescent pattern displayed the 888 numbers in cyan color (PL). (ii) The luminescent pattern showed the 969 digits in green phosphorescence (RTP) when the 365 nm light excitation was turned off. (iii) When a circular polarizer was placed between the pattern and detector (naked eyes), the pattern displayed the digital numbers 354 (CPL) under 365 nm light irradiation. The results show multilevel data encryption and decryption that differentiates from conventional approaches.

### Lig⊂CNC chemical stability and biodegradation

The excited triple state is susceptible to external stimuli, and achieving stable phosphorescence is key to RTP materials. To evaluate the long-term stability of Lig⊂CNC and its RTP property, a co-assembled thin film was soaked in acetonitrile (ACN) for five days. As shown in Fig. 6a, b, the Lig⊂CNC thin film was remarkably stable, and RTP intensity remained unchanged after 1, 2, 3, 4, and 5 days. Meanwhile, the RTP decay at 520 nm of Lig⊂CNC thin film had a stable lifetime, ranging from 106 to 111 ms (Fig. 6c, d). Furthermore, the Lig⊂CNC thin

film exhibited a similar appearance as far as color, luminescence, and phosphorescence (Fig. 6e). Lig⊂CNC thin films that were immersed in other solvents (epichlorohydrin, dichloromethane, ethyl acetate, and tetrahydrofuran) showed no significant reduction in intensity nor lifetime (Supplementary Fig. 18), confirming the excellent chemical stability of Lig⊂CNC. Yet, to evaluate the sustainability of Lig⊂CNC, a typical inorganic RTP material ($Gd_3Al_2Ga_3O_{12}$:$Ce^{3+}$), carbon dot-based RTP material (a-CDs/BA) and natural wood-derived RTP material (C-wood) were used for comparison as far as carbon footprint during the preparation[30,57,58]. The result showed that the carbon dioxide emission of Lig⊂CNC was 3.63 kg while those of C-wood, a-CDs/BA and $Gd_3Al_2Ga_3O_{12}$:$Ce^{3+}$ were 11.98, 31.27 and 133.75 kg, respectively. Thus, the preparation of Lig⊂CNC is environmentally competitive, Fig. 6f. Remarkably, the Lig⊂CNC thin films biodegraded rapidly (16 days) in soil burial conditions, which is in stark contrast to the results obtained from a typical substrate used in optical films, such as polyethylene (PE) (Fig. 6g). To further establish the biodegradation Lig⊂CNC, we tried a series of experiments to test the solubility in water. The Lig⊂CNC thin films displayed no significant changes (16 days) in 30 %RH air or water (Supplementary Fig. 19). This observation suggests the degradation of Lig⊂CNC films in soil burial conditions but points to resistance in humid or water conditions. Based on these findings, a chemically stable yet environmentally-friendly chiroptical film was achieved using all-biobased materials.

### Discussion

In summary, we have developed an all-biobased (CNC and a lig-nosulfonate, Lig) chiroptical film that generates circularly polarized room-temperature phosphorescence. Lignosulfonates allow a green

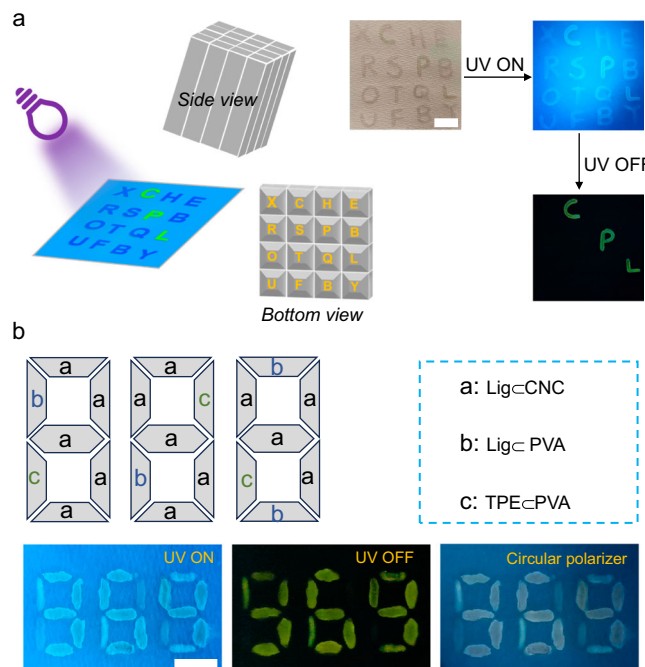

**Fig. 5 | Experiment with an anticounterfeiting device. a** Imprint lithography for a multilevel anticounterfeiting pattern. By turning UV light on/off, different luminescent colors can be recorded. A set of letters except for C, P, and L were printed on Lig⊂CNC using CNC, scale bar = 1 cm. **b** Pattern designed with different phosphors. The luminescence under (left) and without (middle) UV irradiation, and the luminescence through circular polarized filter, scale bar = 1 cm. Note: The CPL firstly went through λ/4 wave plate and then polarizer (PVA polyvinyl alcohol; TPE tetraphenylethene).

phosphorescence at 520 nm with 103.38 ms. Benefiting from the chirality of cellulose nanocrystals, CNC ($g_{abs}$ = 0.9), the co-assembled Lig⊂CNC thin film forms a highly ordered chiral assembly, CP-RTP, with $|g_{lum}|$ = 0.21, much higher than that for the particles suspended in water ($10^{-4}$–$10^{-2}$). Bragg reflection and angle-dependent light intensity measurements rule out the possibility that the CP-RTP of Lig⊂CNC originated from selected transmission through photonic band gap. The co-assembled chiral Lig⊂CNC was readily biodegradable (16 days in soil condition) but resisted typical organic solvents (acetonitrile, epichlorohydrin, dichloromethane, ethyl acetate, and tetrahydrofuran), with no significant changes in RTP intensity and lifetime. Overall, the introduced bio-based chiral co-assembly is shown for its performance (high-dissymmetry circularly polarized phosphorescence) while being cost-effective, biodegradable and environmentally friendly.

## Methods
### Materials
Lignosulfonate (average $M_w$ ~ 52,000) and microcrystalline cellulose (particle size: 250 μm) were purchased from Sigma-Aldrich (Shanghai, China). Acetonitrile (99.5%), epichlorohydrin (99.5%), dichloromethane (99.5%), ethyl acetate (99.5%), tetrahydrofuran (99.5%) and polystyrene (average $M_w$ ~ 400,000) were purchased from Aladdin (Shanghai, China). polyvinyl alcohol (DP = 1750 ± 50, 97%) was purchased from Macklin (Shanghai, China). Deionized water was produced using a Smart-RO ultrapure water system (Hitech Instruments, Shanghai, China). Analytical-grade sulfuric acid (98%) was sulfuric acid were purchased from Kermel Chemical Industry (Tianjin, China).

### Preparation of CNC
Firstly, the microcrystalline cellulose powder (16 g) was mixed with sulfuric acid (140 mL, 64%) under vigorous stirring at 45 °C for 30 min. Then, the suspension was diluted using 1400 ml deionized water to stop the hydrolysis. After standing for 12 h storage, the obtained white bottom suspension was centrifugally washed 3-4 times until the suspension was not stratified and placed in a dialysis membrane (8000-14000 molecular weight cutoff) against deionized water for 7 days. The system was dialyzed until the pH of the dialysate was neutral (the deionized water was changed twice a day). Water was removed from the CNC suspension by using a rotary evaporator until reaching a solids content of 3 wt%. Finally, the CNC suspension was subjected to ultrasonication (400 W, 60% output power) with an ice bath in an ultrasonic cell for 2 minutes. The resultant CNC suspension was stored in a refrigerator until use.

### Preparation of Lig⊂CNC films
A volume of 0, 25, 50, 100, 200 μL lignosulfonate solution (6 mg/mL) was added to 2 mL CNC suspension (3%), respectively and subjected to ultrasonication for 1 min to obtain uniform suspensions. The latter were then placed in plastic Petri dishes (20 mm × 20 mm) and dried in an oven at 30 °C to obtain Lig⊂CNC films with the given lignosulfonate content.

### Characterization
Fluorescence spectra were recorded using an LS-55 fluorescence spectrophotometer (PerkinElmer, Inc., Waltham, MA, USA), equipped with a 120 W Xenon lamp as excitation source. Humidity-dependent afterglow spectra, excitation spectra at different humidity and life-time decay curves were recorded using an FLS1000 photoluminescence spectrometer (Edinburgh Instruments, Livingston, UK), equipped with a Xenon lamp and a microsecond flashlamp (detector: photomultiplier tube, 200 nm <λ< 1700 nm). The chemical changes of the samples were analyzed using FTIR (Nicolet iN10) at the wavelength range between 400 and 4,000 $cm^{-1}$. SEM images were acquired using an FEI Sirion 200 scanning electron microscope (Philips Research, Eindhoven, the Netherlands). CD spectra were recorded using Chirascan spectrometer at 25 °C. CPL spectra were acquired using the JASCO CPL-200 spectrofluoropolarimeter. UV-vis absorption spectra were recorded using a TU-1901 UV-vis double-beam spectrophotometer (Persee General Instrument Co., Ltd., Beijing, China). All performed RTP measurements were conducted in an ambient environment.

**Life-Cycle Assessment (LCA).** The system boundary was defined as cradle to gate. The system included the production of raw materials and energy used in all unit processes. Life cycle assessment of the Lig⊂CNC film preparation was demonstrated and compared with those of fluorescent materials (C-wood, a-CDs/BA and $Gd_3Al_2Ga_3O_{12}$:$Ce^{3+}$), using the same amount of material. The functional unit for samples/control samples were 1 kg. The life cycle inventories of Lig⊂CNC preparation were obtained from laboratory data and further supplements by using detailed cross-check with professionals. The life cycle inventories of Lig⊂CNC, C-wood, a-CDs/BA and $Gd_3Al_2Ga_3O_{12}$:$Ce^{3+}$ preparation are presented in Supplementary Table 1-6. Based on the above data, the Intergovernmental Panel on Climate Change (IPCC) 2021 method was applied to assess the environmental impacts global warming. In addition, the environmental impact of Lig⊂CNC, C-wood, a-CDs/BA and $Gd_3Al_2Ga_3O_{12}$:$Ce^{3+}$ were simulated using SimaproTM (v9.4). The analysis followed International Organization for Standardization (ISO) standard 14040 for LCAs and ISO standard 14067 for carbon footprints.

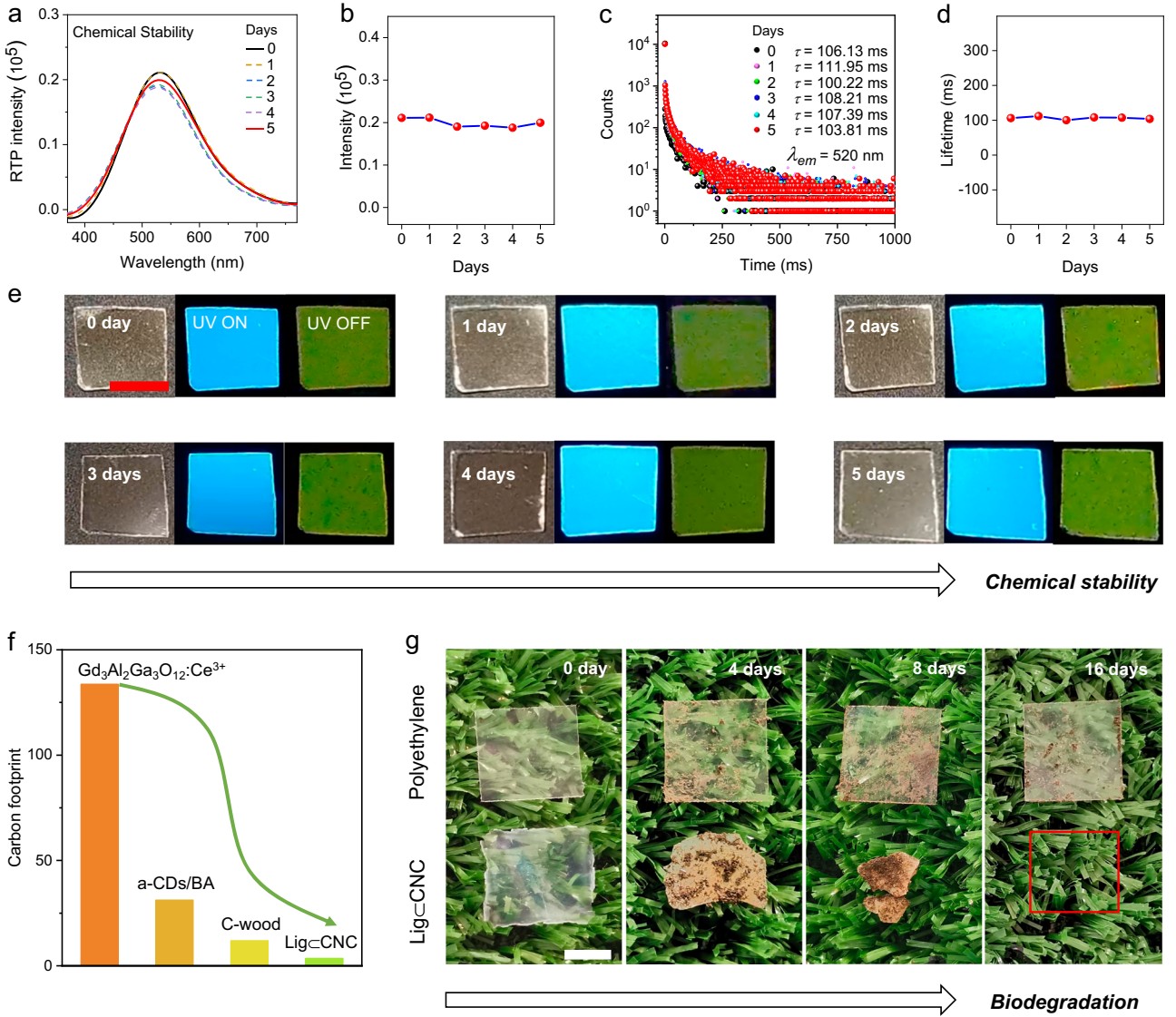

**Fig. 6 | Chemical degradation resistance *versus* biodegradation of the films.**
**a**, **b** RTP intensity changes of Lig⊂CNC soaked in acetonitrile (ACN) solvent for 0 day (black line), 1 day (yellow line), 2 days (blue line), 3 days (green line), 4 days (purple line), 5 days (red line). **c**, **d** Lifetime changes of Lig⊂CNC soaked in ACN solvent for 0 day (black dot), 1 day (purple dot), 2 days (green dot), 3 days (blue dot), 4 days (cyan dot), 5 days (red dot). **e** Image changes of Lig⊂CNC soaked in ACN solvent for 0 to 5 days, scale bar = 1 cm. No significant variation of the property was noted, indicating high chemical stability. **f** Carbon footprint calculated for the preparation of 1 kg of RTP materials. **g** A remarkable biodegradation in soil conditions was shown for Lig⊂CNC but not for polyethylene (PE) films, scale bar = 1 cm.

## Data availability
All relevant data are included in this article and the Supplementary Information file. Source data are available upon request from the corresponding authors. Source data are provided with this paper.

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

## Acknowledgements

Z.J.C wishes to thank the National Natural Science Foundation of China (31890774 and 31800494) and Fundamental Research Funds for the Central Universities (2572023CT06-03). Y.W. wishes to thank the National Natural Science Foundation of China (92356301 and 21908146). O.J.R. wishes to thank the Canada Excellence Research Chair Program (CERC-2018-00006) and the Canada Foundation for Innovation (CFI Project 38623).

## Author contributions

Conceptualization: Z.C., S.L., Y.W., O.J.R.; Methodology: M.C.; Investigation: M.C., Y.R., J.S.; Visualization: M.C., S.L., Z.Q.Y., J.L., S. Liu; Supervision: Z.C., S.L., Y.W., O.J.R.; Writing-original draft: All authors; Writing-review & editing: All authors.

## Competing interests

The authors declare no competing interests.
