## [Peer Review File · Nature Communications]

Biobased and biodegradable films exhibiting circularly polarized room temperature phosphorescenceREVIEWER COMMENTS

Reviewer #1 (Remarks to the Author):

Mengnan Cao and coworkers developed a bio-based thin film showing circularly polarized luminescence (CPL) with room temperature phosphorescence. Here chiral cellulose nanocrystals (CNC) co-assembled with RTP active lignosulfonate biomolecules to get Lig@CNC film which is environmentally friendly, biodegradable (within 16 days), and show CP-RTP with dissymmetry factor of 0.21 and phosphorescence lifetime of 103 ms with a green afterglow of 1s. The authors showed a multilevel data encryption which can find application in the field of anticounterfeiting. Though the work is an extension of their two previous works, Cell Rep. Phys. Sci. 2021, 2, 100542 and Nat. Commun. 2022, 13, 5508, the synthesis, properties, and applications of the films are well explained. This work can be accepted after major revision.

Specific comments

1. "maximum calculated $|g_{lum}|$ value of Lig@CNC reached a value of 0.21, which is considerably higher than that 135 of the CNC thin film ($|g_{lum}| = 0.16$)". No detailed explanation about such an enhancement in $|g_{lum}|$ value.
2. Explain more about the chirality transfer of chiral CNC to achiral lignosulfonates with mechanism.
3. Is it possible to do a temperature-dependent phosphorescence study on Lig@CNC film?
4. Lig@CNC film showing 1s afterglow. What is the afterglow in vacuum condition? Is it enhancing compared to ambient conditions? Is moisture and oxygen affecting the afterglow?
5. Lig@PVA film also shows afterglow. Fig 5b explains that, with a circular polarized filter, only Lig@CNC film is active and can be used for data encryption. Can you provide a CD and CPL spectra of Lig@PVA film as a reference to support the CP-RTP of Lig@CNC film?
6. It is explained that $f_{Lig} > 0.25$ wt% disrupts the co-assembly of the system. Is it possible to provide SEM image of the film with a higher f_{Lig} to give more clarity to that?
7. Please check the luminescence dissymmetry factor equation (line number 124) where (L) is missing.

Reviewer #2 (Remarks to the Author):

In this work, authors fabricated CP-RTP materials from CNC and lignosulfonate. The g achieves to 0.21. The results are interesting, and the materials are novel. This work is anticipated to inspire further advancements in the development of CP-RTR, thus I recommend publication after minor revision:

1. The RTP enhancement is ascribed to the hydrogen bonding between CNC and lignin in this work, thus I suggest to investigate the RTP property of lignin assembled with other substates, such as polyvinyl alcohol and polystyrene.
2. If this RTP thin film is sensitive to humidity? If yes, will the RTP be recovered after drying?
3. Writing mistakes in Fig. 6 caption: I guess that LS@CNC should be Lig@CNC.

4. The authors mentioned the CNC film without lignin exhibits RTP emission with a short lifetime. What is the reason for such phenomenon?
5. Why did the authors use lignosulfonate, is there any special reasons? The authors should give more discussion?

Reviewer #3 (Remarks to the Author):

Circularly polarized luminescence (CPL) materials have garnered significant attention as emerging optical functional materials due to their exceptional optical sensitivity and spatial resolution capabilities. However, most of CPL-active materials are based on fluorescent systems developed through molecular engineering and supramolecular assembly. Consequently, achieving circularly polarized room-temperature phosphorescence (CP-RTP) remains a formidable challenge. In this work, Orlando and co-workers present a coassembled material (Lig⊂CNC) comprising biobased lignin and cellulose nanocrystals, which exhibits CP-RTP emission with a dissymmetry factor of 0.21 and a phosphorescence lifetime of 103 ms. Notably, the Lig⊂CNC film is fully biodegradable and demonstrates remarkable stability even in extreme chemical environments. This successful development of a biobased, environmentally-friendly CP-RTP system offers a new approach to high-performance CPL without the need for complex synthesis and assembly procedures. Consequently, I recommend the publication of this work in Nature Communications after revision. Other comments are as follows:

1. A control experiment should be conducted to validate that the CPL originated from selective transmission, as suggested by Figure 4b.
2. It is necessary to include the luminescence quantum yields of CNC and lig⊂CNC in the manuscript.
3. While the luminescence at 520 nm confirms the phosphorescence of CNC and Lig⊂CNC, it is essential to clarify whether the initial luminescence at 480 nm is also attributed to phosphorescence.
4. To establish the biodegradation of CNC and Lig⊂CNC, it is recommended to provide additional experimental details rather than solely relying on their dissolution in water.

RESPONSE TO THE REVIEWERS' COMMENTS

REVIEWER 1: Mengnan Cao and coworkers developed a bio-based thin film showing circularly polarized luminescence (CPL) with room temperature phosphorescence. Here chiral cellulose nanocrystals (CNC) co-assembled with RTP active lignosulfonate biomolecules to get Lig⊂CNC film which is environmentally friendly, biodegradable (within 16 days), and show CP-RTP with dissymmetry factor of 0.21 and phosphorescence lifetime of 103 ms with a green afterglow of 1s. The authors showed a multilevel data encryption which can find application in the field of anticounterfeiting. Though the work is an extension of their two previous works, Cell Rep. Phys. Sci. 2021, 2, 100542 and Nat. Commun. 2022, 13, 5508, the synthesis, properties, and applications of the films are well explained. This work can be accepted after major revision.

AUTHORS: We sincerely thank the reviewer for the positive comments and suggestion of a revision. We also greatly appreciate the reviewer for the thorough review.

1. “maximum calculated $|g_{lum}|$ value of Lig⊂CNC reached a value of 0.21, which is considerably higher than that 135 of the CNC thin film ($|g_{lum}| = 0.16$)”. No detailed explanation about such an enhancement in g_{lum} value.

AUTHORS: This is a good point that we are happy to address given the noted CPL amplification from CNC to Lig⊂CNC. We found that lignin absorption can match CNC luminescence (Supplementary Fig. 13) and conducted two additional luminescence decay experiments using CNC and Lig⊂CNC. As shown in Supplementary Fig. 14, the luminescence lifetime of pure CNC at 440 nm was calculated to be 80.19 ms, with a luminescence quantum yield of 0.62%. Meanwhile, the lifetime of co-assembled Lig⊂CNC was shortened to 49.91 ms, and the luminescence quantum yield was 3.75%. The evidence suggests the effect of a Förster resonance energy transfer (FRET) process from CNC to lignin within the intermolecular chiral co-assembly system. The FRET efficiency was calculated to be 37.8% (Ref. 53 I. Medintz and N. Hildebrandt, FRET-Förster resonance energy transfer, 2014 Wiley-VCH, Weinheim).

Hence, the CPL $|g_{lum}|$ amplification (0.16 → 0.21) from CNC to Lig⊂CNC is explained as also noted in previous work indicating that FRET can amplify the CPL dissymmetry (Ref. 54 Nat. Commun. 2017, 8, 15727; Ref. 55 Angew. Chem. Int. Ed. 2021, 60, 24549-24557; Angew. Chem. Int. Ed. 2021, 60, 222; Ref. 56 JACS 2022, 144, 5389-5399).

Supplementary Fig. 13 UV-vis absorption spectra of lignin and luminescence spectra of CNC.

Supplementary Fig. 14 Luminescence lifetimes of CNC and Lig \subset CNC ($\lambda_{em} = 440\text{ nm}$) under 365 nm excitation.

2. Explain more about the chirality transfer of chiral CNC to achiral lignosulfonates with mechanism.

AUTHORS: Firstly, we verified the presence of hydrogen bonding between CNC and lignin within the chiral co-assembled system (see Supplementary Fig. 5, see below). Moreover, the luminescence decay indicates the effect of FRET between CNC and lignin (Supplementary Fig. 14, see our reply to comment 1 above). Accordingly, intermolecular CPL transmission takes place by FRET within the chiral co-assembled system, Lig \subset CNC (see also Ref. 54 Nat. Commun. 2017, 8, 15727; Ref. 55 Angew. Chem. Int. Ed. 2021, 60, 24549-24557; Angew. Chem. Int. Ed. 2021, 60, 222; Ref. 56 JACS 2022, 144, 5389-5399).

Supplementary Figure. 5 FTIR spectra of lignosulfonate, CNC and Lig=CNC.

3. Is it possible to do a temperature-dependent phosphorescence study on Lig=CNC film?

AUTHORS: Thanks for the constructive suggestion. The luminescence lifetime of Lig=CNC decreased tracked with the increased temperature, from 77 K to 300 K, in keeping with the expected reduction in phosphorescence at elevated temperatures given the increased non-radiative decay. We added this result as Supplementary Figure. 7. The observations suggest that the emission peak at 520 nm can be attributed to phosphorescence, rather than to thermally activated delayed fluorescence.

Supplementary Fig. 7 Temperature-dependent luminescence lifetime of Lig=CNC at 520 nm.

4. Lig=CNC film showing 1s afterglow. What is the afterglow in vacuum condition? Is it enhancing compared to ambient conditions? Is moisture and oxygen affecting the afterglow?

AUTHORS: The reviewer raises a valuable point that we are happy to address. As stated, we performed RTP measurements under ambient environment but conducted additional experiments to test the effect of humidity and oxygen. As the humidity gradually increased, from 10% to 90%, we noted a sharp reduction of Lig=CNC phosphorescence lifetime (at 520

nm), from ~ 102.28 to ~ 22.82 ms (Supplementary Fig. 8). In addition, we observed that the phosphorescence of LigCNC was sensitive to O_2 , with the lifetime at 520 nm dropping to ~ 45.61 ms under 90% fraction O_2 environment (Supplementary Fig. 10). Such effects are now highlighted in the revised manuscript.

Supplementary Fig. 8 Luminescence lifetime of LigCNC at 520 nm under different humidity environments.

Supplementary Fig. 10 Luminescence lifetime of LigCNC at 520 nm under 90% fraction of O_2 environments.

5. LigPVA film also shows afterglow. Fig 5b explains that, with a circular polarized filter, only LigCNC film is active and can be used for data encryption. Can you provide a CD and CPL spectra of LigPVA film as a reference to support the CP-RTP of LigCNC film?

AUTHORS: We appreciate the opportunity to add CD and CPL spectra in support of our discussion. LigCNC thin film was a chiral co-assembled system; thus, the CD and CPL were observed (Fig. 3). By contrast, chiroptical properties (CD and CPL) were absent. Supplementary Fig. 17 depict nearly null CD and CPL signals for the LigPVA thin film, which confirm that chiral CP-RTP is coming from CNC. Thus, only the LigCNC film is active and, as suggested, can be used for data encryption.

Supplementary Fig. 17 (a) CD and (b) g_{abs} value spectra for Lig_CPVA in the thin-film state; (c) CPL and (d) g_{lum} spectra for Lig_CPVA upon the 365 nm light excitation.

6. It is explained that $f_{\text{Lig}} > 0.25$ wt% disrupts the co-assembly of the system. Is it possible to provide SEM image of the film with a higher f_{Lig} to give more clarity to that?

AUTHORS: Thanks for the valuable suggestion. Accordingly, we performed SEM imaging of co-assembled Lig_CCNC thin films (1% and 2% lignin fractions). Compared to Fig. 3a, the chiral coassembly was destroyed by excess lignin (Supplementary Fig. 12), thus we suggest that 0.25 wt% lignin is an appropriate loading to maintain a chiral coassembly in Lig_CCNC.

Supplementary Fig. 12 Scanning electron microscopy (SEM) images of (a) Lig_CCNC film ($f_{\text{Lig}} = 1\%$) and (b) Lig_CCNC film ($f_{\text{Lig}} = 2\%$).

7. Please check the luminescence dissymmetry factor equation (line number 124) where (L) is missing.

AUTHORS: The reviewer is right and we apologize for our oversight. We have corrected this

point throughout the manuscript.

REVIEWER 2: In this work, authors fabricated CP-RTP materials from CNC and lignosulfonate. The g achieves to 0.21. The results are interesting, and the materials are novel. This work is anticipated to inspire further advancements in the development of CP-RTR, thus I recommend publication after minor revision:

AUTHORS: We greatly appreciate the reviewer for the positive comments and recommendation.

1. The RTP enhancement is ascribed to the hydrogen bonding between CNC and lignin in this work, thus I suggest to investigate the RTP property of lignin assembled with other substrates, such as polyvinyl alcohol and polystyrene.

AUTHORS: This is a very good point, which we address in the revised manuscript to highlight the merits of the LigCNC coassembly relative to other constructs. Hence, to further confirm the effect of hydrogen bonding in RTP enhancing, two different substrates were used, polyvinyl alcohol (PVA) and polystyrene (PS). As shown in **Supplementary Fig. 6**, LigPS, free of hydrogen bonding, displayed extremely weak phosphorescence, with 2.95 ms lifetime. By contrast, the co-assembled LigPVA, which is hydrogen-bonded, clearly showed enhanced phosphorescence, with 168.63 ms. The results strongly support our hypothesis that hydrogen bonding between CNC and lignin enhances the RTP property in the LigCNC system.

Supplementary Figure. 6 a) Phosphorescence spectra and b) lifetime of a Lig PS coassembly. c) Phosphorescence spectra and d) lifetime of LigPVA.

2. If this RTP thin film is sensitive to humidity? If yes, will the RTP be recovered after drying?

AUTHORS: Thank the reviewer for the valuable comment, also reflecting the issue raised by

reviewer 1 (#4). The afterglow RTP thin film was found to be humidity-sensitive, with a reduction of phosphorescence lifetime when the humidity was increased. In particular, the phosphorescence lifetime dropped rapidly from ~ 102.28 ms to ~ 22.82 ms when the humidity increased from 10% to 90% (Supplementary Fig 8). More interestingly, to the point raised by Reviewer 2, the quenched lifetime was fully recovered after drying the LigCNC coassembly. The “humidity-drying” RTP lifetime was stable and was recovered during wetting-drying cycles, as shown after more than four cycles (Supplementary Fig 9).

Supplementary Fig. 8 Luminescence lifetime of LigCNC at 520 nm under different humidity environments.

Supplementary Fig. 9 Lifetime of LigCNC upon recyclable treatment of humidity (90 %RH) and drying (20 %RH).

3. Writing mistakes in Fig. 6 caption: I guess that LS=CNC should be LigCNC.

AUTHORS: Thanks for pointing this out. This is now corrected

4. The authors mentioned the CNC film without lignin exhibits RTP emission with a short

lifetime. What is the reason for such phenomenon?

AUTHORS: In contrast to conventional organic luminescent materials, characterized by the presence of aromatic rings and double bonds, cellulose does not have the traditional valence conjugation. Also, there are heteroatoms with lone pair electrons in cellulose. Hence, cellulose have been found to exhibit photoluminescent properties due to the presence of nonconventional luminophores in their molecular structure. The photoluminescence arising from nonconventional luminophores is termed clustering-triggered emission. However, the latter only emit light within the blue-light region, with low photoluminescence quantum yield. (see Nat. Commun. 2023. 14, 409; Mater. Chem. Front. 2021. 5, 6693-6717; Chin. J. Polym. Sci. 2019. 37, 409-415; Chem. Soc. Rev. 2021. 50, 12616-12655)

5. Why did the authors use liginosulfonate, is there any special reasons? The authors should give more discussion?

AUTHORS: Two important reasons for the choice of lignin type includes the commercial availability and the solubility. Liginosulfonates are lignin derivatives that are widely available, commercially, and are currently used in the oil and gas sector, as dispersant and in crop protection as well as other industries. By contrast, 96% of the most typical lignins (Kraft lignins, for example), are incinerated to supply heat and energy (Chem. Mater. 2020. 32, 4324–4330). With a better understanding of the nature and chemistry of liginosulfonates, valuable materials, such as aromatic chemicals depolymerized from lignin and polymer composites are being developed (Chem. Rev. 2018. 118, 614–678). From a chemical perspective, lignin consists of p-hydroxyphenyl (H), guaiacyl (G) and syringyl (S) units, linked by β -O-4 and C-C bonds. (ACS Catal. 2018. 8, 1614–1620). These aromatic units (G, S and H) endow liginosulfonate with great potential as a RTP chromophore. Additionally, the alcohol hydroxyl and phenolic hydroxyl groups in lignin contribute to the chemical or physical interactions with rigid matrices. These properties have enabled liginosulfonates to become an ideal aromatic component for RTP emission materials (see Nat. Rev. Chem. 2023 7, 800-812; Green Chem. 2023. 25, 1406–1416). Moreover, liginosulfonates are easily dispersible in water, contrary to Kraft lignins, and can be conveniently coassembled with CNC via hydrogen bonding (Supplementary Fig. 5, below), which is beneficial to RTP enhancement (Fig. 2a, below). Furthermore, liginosulfonates are biocompatible and show low toxicity, good processability, and facile preparation.

Supplementary Figure. 5 FTIR spectra of liginosulfonate, CNC and Lig-CNC.

Figure 2 (a) Photos showing time-dependent photoluminescence of assembled CNC and co-assembled LigCNC.

REVIEWER 3: Circularly polarized luminescence (CPL) materials have garnered significant attention as emerging optical functional materials due to their exceptional optical sensitivity and spatial resolution capabilities. However, most of CPL-active materials are based on fluorescent systems developed through molecular engineering and supramolecular assembly. Consequently, achieving circularly polarized room-temperature phosphorescence (CP-RTP) remains a formidable challenge. In this work, Orlando and co-workers present a coassembled material (LigCNC) comprising biobased lignin and cellulose nanocrystals, which exhibits CP-RTP emission with a dissymmetry factor of 0.21 and a phosphorescence lifetime of 103 ms. Notably, the LigCNC film is fully biodegradable and demonstrates remarkable stability even in extreme chemical environments. This successful development of a biobased, environmentally-friendly CP-RTP system offers a new approach to high-performance CPL without the need for complex synthesis and assembly procedures. Consequently, I recommend the publication of this work in Nature Communications after revision. Other comments are as follows:

AUTHORS: We appreciate the recommendation provided by the reviewer and the comments on the successful development of an environmentally-friendly CP-RTP system.

1. A control experiment should be conducted to validate that the CPL originated from selective transmission, as suggested by Figure 4b.

AUTHORS: Thanks for the constructive advice. According to the optical setup shown in Fig. 4a, we conducted a selective transmission test for chiral liquid crystal (2.33 w% R5011 in TEB300) as a reference, which was found to possess reflection band from 500 to 550 nm (Supplementary Fig. 15). As shown in Supplementary Fig. 16, the laser characteristic at 532 nm was evaluated optically by analyzing the azimuthal angle-dependent transmission with a round curve, demonstrating a typical selective transmission. Meanwhile, the “8” profile of LigCNC ruled out selected transmission/reflection-dependent CPL (Fig 4b).

Supplementary Fig. 15 Reflection of chiral liquid crystal assembly (2.33 w% R5011 in TEB300) indicating strong reflection ranging from 500 to 550 nm

Supplementary Fig. 16 Transmission intensity of linearly polarized light at 532 nm across a chiral LC system (2.33 w% R5011 in TEB300).

2. It is necessary to include the luminescence quantum yields of CNC and LigCNC in the manuscript.

AUTHORS: Thanks for the suggestion. Accordingly, the quantum yield of CNC and LigCNC are reported to be 0.62% and 3.75%, respectively (lines 16-17, page 7).

3. While the luminescence at 520 nm confirms the phosphorescence of CNC and LigCNC, it is essential to clarify whether the initial luminescence at 480 nm is also attributed to phosphorescence.

AUTHORS: This is a good point. As shown in **Supplementary Fig. 3**, the luminescence lifetime of LigCNC was calculated as 75.22 ms at 480 nm. This observation suggests that the initial

luminescence at 480 nm can be attributed to phosphorescence.

Supplementary Fig. 3 Luminescence lifetime of LigCNC ($\lambda_{em} = 480$ nm, 298 K).

4. To establish the biodegradation of CNC and LigCNC, it is recommended to provide additional experimental details rather than solely relying on their dissolution in water.

AUTHORS: Thanks for the opportunity to demonstrate the biodegradation prospects. We carried out a series of experiments to test the solubility in water and degradation in humid environments. The main conclusion is that the LigCNC thin films show no significant changes (16 days in 30% RH or water, **Supplementary Fig. 19**). This contrasts to results obtained in soil burial experiments. Overall, the observations suggest that the degradation of LigCNC films is due to enzyme and microorganism activities, as shown soil burial tests.

Supplementary Fig. 19 Solubility of LigCNC film in 30 %RH or water.

REVIEWERS' COMMENTS

Reviewer #1 (Remarks to the Author):

All the comments are addressed satisfactorily and the revised manuscript can be accepted.

Reviewer #2 (Remarks to the Author):

The manuscript has been revised seriously by authors. It can be accepted.

Reviewer #3 (Remarks to the Author):

This work has been greatly improved after revision, and the authors have addressed my comments. Thus, the manuscript can be published as is.